# Update of the Seismogenic Potential of the Upper Rhine Graben Southern Region

Sylvain Michel[1,2], Clara Duverger[2], Laurent Bollinger[2], Jorge Jara[1], Romain Jolivet[1,3]

[1] Laboratoire de Géologie, Département de Géosciences, Ecole Normale Supérieure, PSL Université, CNRS UMR 8538, 24 Rue Lhomond, 75005, Paris, France.
[2] CEA, DAM, DIF, F-91297 Arpajon, France
[3] Institut Universitaire de France, 1 rue Descartes, 75005, Paris

*Correspondence to*: Sylvain Michel (sylvain_michel@live.fr)

**Abstract.**

The Upper Rhine Graben (URG), located in France and Germany, is bordered by north-south trending faults, some of which are considered active, posing a potential threat to the dense population and infrastructures on the Alsace plain. The largest historical earthquake in the region was the M6.5+/-0.5 Basel earthquake in 1356. Current seismicity (M>2.5 since 1960) is mostly diffuse and located within the graben. We build upon previous seismic hazard studies of the URG by exploring uncertainties in greater detail and revisiting a number of assumptions. We first take into account the limited evidence of neotectonic activity, then explore tectonic scenarios that have not been taken into account previously, exploring uncertainties for $M_{max}$, its recurrence time, the *b*-value, and the moment released aseismically or through aftershocks. Uncertainties on faults' moment deficit rates, on the observed seismic events' magnitude-frequency distribution, and on the moment-area scaling law of earthquakes are also explored. Assuming a purely dip-slip / normal faulting mechanism associated to a simplified 3 main fault model, $M_{max}$ maximum probability is estimated at $M_w6.1$. Considering this scenario, there would be a 99% probability that $M_{max}$ is less than 7.3. In contrast, with a strike slip assumption associated to a 4 main fault model, consistent with recent paleoseismological studies and the present-day stress field, $M_{max}$ is estimated at $M_w6.8$. Based on this scenario, there would be a 99% probability that $M_{max}$ is less than 7.6.

# 1    INTRODUCTION

The Upper Rhine Graben (URG), located in France and Germany, is bounded by north-south trending faults, some of which are considered active, posing a potential threat to the dense population and the industrial and communication infrastructures of the Alsace plain (Figure 1). The largest historical earthquake in the region was the 1356 Basel earthquake with a maximum intensity equal to or greater than IX (Mayer-Rosa and Cadiot, 1979; Fäh et al., 2009), an earthquake presently associated to a magnitude between M6.5+/-0.5 (Manchuel et al., 2017) and M6.9+/-0.2 (Fäh et al., 2009). Current seismicity (M>2.5 since 1960) is mostly diffuse and located within the graben (Doubre et al., 2022), hence the difficulty to attribute individual events to a given fault segment. The bordering faults themselves are relatively quiet except for the south-eastern section of the graben, near Mulhouse-Basel, where natural seismic sequences (Rouland et al., 1983; Bonjer, 1997) and induced seismicity (Kraft and Deichmann, 2014) have been observed. Seismic activity actually varies along the URG with an increasing rate of events towards the south (Barth et al., 2015). The relative rate between small and large events (b-value from the Gutenberg-Richter law) also increases towards the south indicating a surplus of small earthquakes or a deficit of large events roughly south of Strasbourg (Barth et al., 2015). Focal mechanisms of earthquakes suggest that the region is subject to strike-slip regime with some normal component (Mazzotti et al., 2021), consistent with the large wavelength strain inferred from geodetic data (Henrion et al., 2020). Characterizing the slip rates of the graben's faults based on geodetic data remains challenging. Indeed regional glacial isostatic adjustments, local subsidence and low tectonic strain rates result in a heterogeneous velocity field with values below 0.2 mm/yr and often within measurement uncertainties  (Fuhrmann et al., 2015; Henrion et al., 2020).

The seismic hazard of the URG has been evaluated by multiple studies at the national/European scale (Grünthal et al., 2018; Drouet et al., 2020; Danciu et al., 2021).  Furthermore, the seismic hazard of the southern region of the URG in particular has recently been assessed by Chartier et al. (2017) with a focus on the Fessenheim nuclear power plant (Figure 1). This study evaluates the seismic hazard using a fault-based approach, taking into account the network of potentially active faults characterized by Jomard et al. (2017). This fault-based work involves a moment budget approach, which involves comparing the rate of moment release by seismicity and the rate of moment deficit (MDR) accumulating along locked portions of faults between large earthquakes (i.e. the tectonic loading rate of each fault). Since the period of seismological observation (a few centuries) is too short to be representative of the long-term behavior of seismicity, Chartier et al. (2017) built instead a seismicity model assumed to be representative of the long-term Magnitude-Frequency Distribution (MFD) of earthquakes, a method similarly used in former studies (e.g. Molnar, 1979; Anderson and Luco, 1983; Avouac, 2015). Earthquakes below

$M_w5$ are disregarded (Bommer and Crowley, 2017; Chartier et al., 2017). Earthquakes between $M_w5$ and 6 are
assumed to follow the MFD of the catalog of earthquakes they consider. This catalog integrates several sources of
instrumental and historical earthquakes including sources from the *Laboratoire de Détection et de Géophysique*
of the *Commissariat à l'Énergie Atomique et aux énergies alternatives* (CEA-LDG; http://www-dase.cea.fr/) and
from the FPEC (French Parametric Earthquake Catalogue; Baumont and Scotti, 2011), the IRSN contribution to
SHEEC (SHARE European Earthquake Catalogue; Stucchi et al., 2013). MFDs are estimated based on a French
seismotectonic zoning scheme defined by Baize et al. (2013). Earthquakes with magnitude above $M_w6$ are assumed
to occur on the fault planes (Jomard et al., 2017). Chartier et al. (2017) consider two types of model: (1) Each fault
ruptures only as its maximum magnitude event, which is controlled by the surface area of the seismogenic fault
segment (characteristic earthquake model); (2) Events follow the Gutenberg-Richter (GR) law with a b-value equal
to 1, and the maximum magnitude, $M_{max}$, is fixed as in the previous model. The recurrence times of the $M_w>6$
events are then calibrated so that the rate of moment released by the seismicity models matches the MDR estimated
from neotectonic data (Chartier et al., 2017; Jomard et al., 2017). The authors explore different fault geometries
(e.g. dip and seismogenic depth) using a logic-tree methodology and then proceed to the Probabilistic Seismic
Hazard Assessment (PSHA) of the region, providing a map of the probability of exceedance of Peak Ground
Acceleration (PGA) within a time period.
A number of strong assumptions are made within this framework. As mentioned previously, a simplified fault
network is used (Jomard et al., 2017), which constrains the seismogenic area available for ruptures. Expert choices
have also been made to distribute slip rates (i.e. loading rates) originally attributed to faults that have been removed
from the initial fault network (Nivière et al., 2008) on other fault segments. On a number of faults, no estimates of
neotectonic slip rate are available (e.g. West Rhenish Fault) and the authors have chosen to apply slip rates
equivalent to those from other nearby faults (0.01 to 0.05 mm/yr). The neotectonic data are actually only along-
dip slip rate estimates. No along-strike slip rates have yet been published due to the lack of markers to quantify
horizontal offsets along faults and this component has thus been ignored. In addition, Chartier et al. (2017) do not
consider continuous probabilities as they apply a logic-tree method. Chartier et al (2017) fix the b-value to 1,
choose the seismogenic depth to be either 15 or 20 km and do not take into account multi-segment ruptures when
estimating a $M_{max}$ for each fault segment.
In this study, we build upon Chartier et al. (2017) seismic hazard evaluation of the southern URG by exploring
uncertainties in greater detail, revisiting a number of assumptions. We use the methodology from Rollins and
Avouac (2019) and Michel et al. (2021), which allows to evaluate the seismogenic potential of faults in a
probabilistic fashion and explore uncertainties for parameters such as the b-value or $M_{max}$. We use the fault
network and slip rates taken into account by Nivière et al. (2008), disregarding the Western Rhenish Fault for
which, to our knowledge, no slip rate data is available. We assume faults can rupture simultaneously (i.e. multi-
segment rupture). In the following sections, we start by describing the concepts and methods we use to constrain
the seismogenic potential of the URG, and then describe the data available before discussing the robustness of our
results.
**2  METHOD**
We use the methodology from Michel et al. (2021) in order to estimate the seismogenic potential of the upper
Rhine Graben, including $M_{max}$ and its recurrence time. As in Chartier et al. (2017), we produce seismicity models
representative of the long-term behavior of earthquakes. We assume that the MFDs of background earthquakes
follow a Gutenberg-Richter power law up to $M_{max}$. We define background earthquakes as mainshocks, as opposed
to their subsequent aftershocks. We assume that their timing of occurrence is random, following a Poisson process.
Each model is controlled by three parameters: (1) $M_{max}$, (2) the recurrence time of events of a certain
magnitude, $\tau_c$, and (3) the b-value. We use two types of model, namely the tapered and truncated models (Rollins
and Avouac, 2019; Michel et al., 2021; Figure S1). The tapered model type assumes a non-cumulative power-law
MFD truncated at $M_{max}$, which gives rise to a tapered MFD in the cumulative form (i.e. the traditional display
when representing the Gutenberg-Richter law). The truncated model type assumes instead a MFD with a
distribution truncated at $M_{max}$ in the cumulative form.
The seismicity models are then tested against three constraints: (1) the moment budget, as in Chartier et al. (2017),
which implies that moment released by slip on the fault should match the moment deficit accumulating between
earthquakes over a long period of time; (2) the moment-area scaling law, an empirical scaling law relating rupture
area to slip for each earthquake, and (3) the MFD of observed seismicity. Each of these constraints are described
in more detail in the following sub-sections. The data and associated uncertainties used for the constraints are
discussed in the following section (i.e. Section 3).
**2.1  Moment budget**
A moment budget consists in comparing the rate of moment released from slip events (seismic or aseismic),
$\dot{m}_0^{Total}$, with the moment deficit rate, $\dot{m}_0^{def}$, accumulating between slip events. The moment deficit rate is defined
by the equation $\dot{m}_0^{def} = \int \mu \ \dot{D}^{def} \ dA$, where $\mu$ is the shear modulus, $A$ is the area that remains locked during the
interseismic period (i.e. the potential seismogenic zone), and $\dot{D}^{def}$ is the rate at which slip deficit builds up. Since
the distribution of locked segments of faults and their associated loading rates cannot yet be determined for the
URG from geodetic measurements, $A$ is assumed to be homogeneous along-strike for each fault, while we consider
possible the seismogenic width to change from one fault to another. The rate at which slip deficit builds up, $\dot{D}^{def}$,
is evaluated based on neotectonic information (see Section 3.1). The total moment released, $\dot{m}_0^{Total}$ is calculated
based on the rate of moment release of the long-term seismicity model. Since the long-term seismicity model only
considers mainshocks, we included a fourth parameter, $\alpha_s$, that represents the proportion of moment released by
background seismicity (Avouac, 2015), $m_0^{Bckgrd}$, relative to the total moment released (including aftershocks and
aseismic afterslip). If $\dot{m}_0^{def} = \dot{m}_0^{Total} = \dot{m}_0^{Bckgrd} / \alpha_s$, then the moment budget is said to be balanced.
The cumulative MFD for tapered and truncated seismicity models achieving a balanced moment budget have an
analytical form and are a function of $M_{max}$, $b$, $\dot{m}_0^{def}$ and $\alpha_s$ (see Rollins and Avouac, 2019, and references therein).
We can therefore estimate the probability of a seismicity model balancing the moment budget, $P_{Budget}$, by
sampling the *a priori* distributions of those parameters.
**2.2    Moment-area scaling law**
According to global earthquake statistics, the moment released by an earthquake, $m_0^{Seis}$, is proportional to the area
of its rupture, $A_{eq}$, such that $m_0^{seis} \propto A_{eq}^{3/2}$ (Wells and Coppersmith, 1994; Leonard, 2010; Stirling et al., 2013).
We use this scaling to evaluate whether a seismic event of a given magnitude has a rupture area that fits within the
seismogenic zone. By considering the spread of the empirical distribution of magnitude vs. area, we assume the
probability distribution function of an event of magnitude $M_w$ to be probable considering this scaling, $P_{scaling}$. We
use here the self-consistent scaling law, and related uncertainties, as defined by Leonard (2010) in the dip-slip
equation (the strike-slip equation is in any case almost the same).
**2.3    Earthquake catalog**
We test whether the observed MFD from earthquake catalogs may be a sample of the distribution of the long-term
seismicity models we are building. Effectively, we evaluate the likelihood of our observed MFD given the
distribution of the models. Since we only consider mainshocks, we define the likelihood of the observed seismicity
catalog, $P_{Cat}$, as $P_{Cat} = \prod_i P_{poisson}^{M_i}$, where $P_{poisson}^{M_i}$ is the probability to observe $n_{obs}^{M_i}$ events, within the magnitude
bin $M_i$, occurring during the time period $t_{obs}^{M_i}$, assuming the long-term mean recurrence of events is $\tau_{model}^{M_i}$:
$$P_{poisson}^{M_i}\left(n_{obs}^{M_i}, t_{obs}^{M_i}, \tau_{model}^{M_i}\right) = \frac{\left(t_{obs}^{M_i}/\tau_{model}^{M_i}\right)^{n_{obs}^{M_i}}}{\left(n_{obs}^{M_i}\right)!} e^{-t_{obs}^{M_i}/\tau_{model}^{M_i}}.$$
Effectively, for a given seismicity model, we generate randomly 2500 declustered earthquake catalogs. We
evaluate the likelihood of each catalog and define $P_{Cat}$ as the average of these likelihood values.
Note that we follow the recommendation by Felzer (2008) while exploring magnitude uncertainties and correct
the magnitudes of each event by $\Delta M = (b^2\sigma^2)/(2\,log_{10}(e))$, where $b$ is the declustered catalog $b$-value, $\sigma$ is the
standard deviation for the event's magnitude, and $e$ is the exponential constant.
**2.4    Seismicity model probability and marginal probabilities**
Finally, the probability of a seismicity model is defined as $P_{SM} = P_{Budget}\,P_{Cat}\,P_{scaling}$ which depends, among
others, on $M_{max}$ and $b$ (Michel et al., 2021). The evaluation of the parameters to estimate $P_{SM}$ are discussed in
Section 3. Marginal probabilities such as $P_{M_{max}}$, the probability of $M_{max}$, and $P_b$, the probability of the $b$-value,
can be estimated based on $P_{SM}$. We also define $P(\tau_{max}\,|\,M_{max})$ as the probability of the rate of $M_{max}$, and
$P(\tau\,|\,M_w)$ as the probability of the rate of events with magnitude $M_w$, which accounts for all earthquakes from all
of the models (i.e. not only $M_{max}$). Probabilities needed for estimating seismic hazard (e.g. PSHA) such as the
probability to have an event above magnitude $M_w$ for a time period $T$, $P(M > M_w\,|\,T)$, can likewise be evaluated.
**3    DATA AND ASSOCIATED UNCERTAINTIES**
We present in this section the data and their associated uncertainties used to evaluate each constraint. Hereafter,
the $\mathcal{U}$ and $\mathcal{N}$ symbols will stand for uniform and normal distribution, respectively. Table 1 summarizes the
uncertainties taken for each parameter.
**3.1    Neotectonic data, seismogenic along-dip width and moment deficit rate**
In order to evaluate the MDR for the moment budget constraint (Section 2.1), we must infer estimates of loading
rate (i.e. $\dot{D}^{def}$) for each fault taken into account. The slip rate on each fault is taken from Nivière et al. (2008) for
the Rhine River, Black Forest, Weinstetten and Lehen-Schonberg faults (the Landeck or West Renish faults are
not considered). Their slip rates rely on estimates of the cumulative vertical displacement of the faults based on
Pliocene-Quaternary sediments thickness variations measured from 451 boreholes, assuming that the
accommodation space opened by tectonic motion is completely balanced (or over-balanced) by sedimentation.
However, potential erosional periods due to the piracy of the Rhine River might bias the measurements, thus the
values are to be interpreted as maximum displacement estimates. Nivière et al. (2008) inferred vertical slip rates
of 0.07 and 0.17 mm/yr from the age of the sediments for the Rhine River and Weinstetten faults respectively. The
Lehen-Schonberg fault slip rate reaches between 0.04 and 0.1 mm/yr. While borehole observations do not allow
to conclude on the Pliocene-Quaternary slip rate of the Black Forest fault, this structure is suggested to be inactive
during this time period, and that the deformation is now accommodated by the other aforementioned faults (Nivière
et al., 2008). Note that these are vertical slip rate estimates and the along-strike component is for the moment
neglected. For the moment rate calculation, we project vertical slip rates on the along-dip direction considering
the dip angles of each fault.
The seismogenic down-dip extent of a fault depends on the temperature gradient (e.g. Oleskevich et al., 1999),
among other parameters. Indeed, between the isotherms 350°C and 450°C, quartzo-feldspathic rocks undergo a
transition in frictional properties (Blanpied et al., 1995) from a rate-weakening (<350°C), potentially seismogenic
behavior to a rate-strengthening (>450°C), stable sliding behavior (Dieterich, 1979; Ruina, 1983). The geothermal
gradient below the URG is higher than in the surrounding regions due to its tectonic history (Freymark et al.,
2017). Based on borehole temperature measurements from Guillou-Frottier et al. (2013), we estimate the envelopes
of the geothermal gradient in the southern URG (Figure S2), assuming a linear temperature gradient with depth,
and show that the frictional property transition would occur between depths of 6 (shallowest position of the 350°C
isotherm; Figure S2) and 18 km (deepest position of the 450°C isotherm; Figure S2). In this study, we define the
PDF of the seismogenic down-dip extent as a uniform distribution between 0 and 6 km depth associated with a
linear taper down to 18 km. The linearity of the taper implies that the position of the fault's transition to a fully
rate-strengthening behavior (>350-450°C) has a uniform probability to fall between 6 km (shallowest position of
the 350°C isotherm according to Figure S2) and 18 km depth (deepest position of the 450°C isotherm; Figure S2),
i.e. *Rate-Strengthening Transition* $\in \mathcal{U}(6,18)$ km.
Additionally, the southern part of the URG is the site of a potash-salt evaporitic basin (Lutz and Cleintuar, 1999;
Hinsken et al., 2007; Freymark et al., 2017), which reaches a maximum depth of ~2 km. Such formations may not
accumulate any moment deficit as the yield stress of evaporites is very low (Carter and Hansen, 1983). We assume
each fault is potentially impacted by this formation, hence modulating the seismogenic thickness and in turn the
seismogenic area available for a rupture. The resulting PDF for the seismogenic thickness is the convolution of
the PDF of the down-dip extent of the seismogenic zone with the PDF of the evaporitic basin thickness taken as
$\mathcal{U}(0,2)$ km.. Combining both temperature and salt basin assumptions leads to a PDF of the along-dip seismogenic
width, which is uniform down to ~5 km and decreases linearly until ~17 km (Figures S3 to S6).
The moment deficit is then the product of the length of each fault, their seismogenic width, the neo-tectonic long-
term slip rate, and the shear modulus that we fix to 30 GPa (same as in Chartier et al., 2017). Each fault is assumed
to have its own seismogenic width. The moment deficit rate of each fault is shown in Figure 1. The PDFs for each
of the fault's constitutive parameters are shown in Figure S3 to S6. By considering the range of the fault's
geometrical parameters, which considers also the Black Forest Fault even though it is assumed to be non-active,
we obtain the moment-area constraint shown in Figure 2. Events up to $M_w 6.5$ are equiprobable while those
above $M_w 7.7$ are extremely improbable.
**3.2    Instrumental and historical seismicity catalogs**
To constrain the MFD of the long-term seismicity models with an observational seismicity catalog, as described
in Section 2.3, we need to evaluate from the observational catalog the number of events per magnitude bin $n_{obs}^{M_i}$
over a period of time $t_{obs}^{M_i}$ (Section 2.3). We use the earthquake catalog from Drouet et al. (2020). This catalog was
built from multiple former catalogs. It relies mostly on the FCAT-17 catalog (Manchuel et al., 2018), which is
itself a combination of the instrumental catalog SiHex (SIsmicité de l'HEXagone; Cara et al., 2015) for the 1965-
2009 period, and a historical catalog based on the macroseismic database of SISFRANCE (BRGM, IRSN, EDF),
intensity prediction equations from Baumont et al. (2018) and the macroseismic moment magnitude determination
from Traversa et al. (2018) for the 463-1965 period. Events located more than 20 km from the French border, not
provided by the FCAT-17, are based on the SHEEC catalog (Stucchi et al., 2013; Woessner et al., 2015). Finally,
events between 2010 and 2016 come from the CEA-LDG bulletins (https://www-dase.cea.fr). All event
magnitudes are given in $M_w$ and uncertainties are provided. Anthropic events are expected to be already removed
from the catalog (Cara et al., 2015; Manchuel et al., 2018).
We select events within the coordinates [6°, 8.5°] longitude and [47°, 49.5°] latitude, i.e. a broad region covering
the whole URG, and divide the catalog into two time periods, an instrumental period and a historical one taking
events from 1980 onwards and 1850 onwards, respectively. We decluster both catalogs to compare them with the
long-term seismicity models (Section 2.3). Declustering is based on the methodology of Marsan et al. (2017),
which evaluates the probability that an earthquake is a mainshock. Declustering is applied based on a completeness
magnitude, $M_c$, of 2.2 and 3.2 for the instrumental and historical catalogs, respectively (Text S1; Figures S7 and
S8). From the resulting catalogs, we keep events from 1994 onwards and 1860 onwards for the instrumental and
historical catalogs, respectively (Figures S7 and S8), in order to avoid border effects from declustering. For the
instrumental catalog, 1994 is also the date from which the seismicity rate appears relatively constant (Figure S7).
We then select events in the region of interest (i.e. the southern part of the URG), taking into account only
earthquakes located within a 10 km buffer around the faults considered, including the Black Forest fault (Figure
3). Note that since no events below $M_c$ are considered, there is a lack of events which falls in the magnitude bins
directly above $M_c$ while exploring magnitude uncertainties. Thus, when applying the earthquake catalog constraint
(Section 2.3), we take events with $M_w \geq 2.8$ and $M_w \geq 4.3$ for the instrumental and historical catalogs,
respectively (Felzer, 2008) (Figure 3).
**3.3    Constitutive parameters of seismicity models**
As mentioned in Section 2.1, the cumulative MFD for tapered and truncated seismicity models balancing the
moment budget can be defined as a function of $M_{max}$, $b$, $\dot{m}_0^{def}$ and $\alpha_s$. We explore these parameters using a grid
search with $M_{max}$ and $b$ sampled uniformly over $M_{max} \in \mathcal{U}(4.5, 9.9)$ and $b \in \mathcal{U}(0.1, 1.45)$, respectively. Based
on global statistics of the post-seismic response following earthquakes (Alwahedi and Hawthorne, 2019; Churchill
et al., 2022), we assume that the PDF of $\alpha_s$ is a Gaussian distribution with $\mathcal{N}(0.9, 0.25)$ (Figure S9). Finally, the
PDF of the MDR for each fault is assumed to be uniform between 0 and the estimate based on the maximum slip
rate from Nivière et al. (2008) (Section 3.1). We thus include scenarios for which almost no moment deficit
accumulates on the fault (i.e. the fault slips aseismically or accumulates no strain over long periods of time). This
assumption contrasts with the choice made by Chartier et al. (2017) who assume that each fault is fully locked
over a seismogenic width terminating at either 15 or 20 km. Doing so, we explore a broad range of possible models.
**4    RESULTS**
The combination of constraints (Section 2) leads to the results shown in Figure 4. For the truncated model, the
marginal probability of $P_{SM}$ in the $M_{max}$ and $\tau_{max}$ space is represented by the gray shaded distribution in Figure
4 (not shown for the tapered model since the models taper at $M_{max}$). The marginal probability of $M_{max}$ for the
tapered model (in green) peaks at 6.1, while the one for the truncated model (in blue) is bi-modal with peaks at 5.2
and 5.8. For the truncated model (not the tapered model for the same reason as previously indicated), the marginal
probability $P(\tau_{max} \mid M_{max} = 5.8)$ (solid blue line in the y-axis) peaks at ~1000 yrs. Taking $M_{max} =$6.6 or 7.0, a
number close to the estimated magnitude of the 1356 Basel earthquake, the marginal probability would instead
peak at ~16,000 and ~80,000 yrs, respectively.
The marginal probabilities $P(\tau \mid M_w = 6.1)$ and $P(\tau \mid M_w = 5.8)$ for the tapered and truncated models (green and
blue dotted lines on the y-axis, respectively), which take all events from the seismicity models into account (not
only $M_{max}$), have instead peaks at ~16,000 yrs and ~10,000 yrs, respectively. The marginal probability $P_b$ peaks
at ~0.85 and 0.9 for the tapered and truncated models, respectively.
The effect with and without the moment-area scaling law is shown in Figure 5. Adding the scaling law constraint
does not change the mode of $P_{M_{max}}$ but completely rejects scenarios with $M_{max}$>7.8.
Finally, the probabilities $P(M > M_w \mid T)$ for $T = 100$ and 10,000 yrs are also shown in Figure 5. As an example,
the probability of occurrence for an event above $M_w 6.5$ (similar to the 1356 Basel earthquake) for an observational
period of 100 yrs is ~0.1% for both the tapered and truncated models. For an event above $M_w 6.0$ and for the same
period, this probability is  instead ~1% for both models (see zoom in Figure 5.c).
The correlations between $M_{max}$, the moment deficit rate, the $b$-value, and $\alpha_s$, for both the tapered and truncated
models but without the scaling law constraint, are shown in Figures S10 and S11. For both models, probable $M_{max}$
increases with increasing $b$-value (Figure S10.a and S11.a), highlighting strong interdependency between the two
parameters. Raising the moment deficit rate will control the minimum probable $M_{max}$ (Figures S10.b and S11.b)
but will also tend to exclude scenarios with a high b-value (>1.25; Figures S10.f and S11.f). While other trends
are expected between parameters, they seem less visible likely due to the uncertainties of the parameters explored,
and we thus do not pursue further analysis between those parameters.
The results if we combine the PDFs from the tapered and truncated models using a mixture distribution are shown
in Figure S12. $P_{M_{max}}$ has a main peak at 5.9 and a smaller peak at 5.2, which originates from the truncated model.
$P(\tau \mid M_w = 5.9)$ peaks instead at ~13 000 yrs.
**5    DISCUSSION**
**5.1    Sensibility to earthquake catalog declustering**
The catalog declustering (i.e. removal of aftershocks) may have a significant impact on the results (Section 2.3),
influencing the shape of the observed MFD of earthquakes. In this study, we applied the methodology of Marsan
et al. (2017), which is based on the ETAS framework and intrinsically assumes that background events have
Poisson behavior. Other declustering methodologies are available and we test here the one from Zaliapin and Ben-
Zion (2013) based on the nearest-neighbor distances of events in the space-time-energy domain. The results from
this methodology produce background seismicity catalogs with more events than the one from Marsan et al. (2017)
(Text S2 and Figures S13 to S15), but infers larger b-values when combining the instrumental catalog with the
historical one (as inferred by Figure 6.b). The analysis of the seismogenic potential of the URG using Zaliapin and
Ben-Zion (2013) methodology results with $P_{M_{max}}$ peaking at M6.3 for the tapered model, and is still bi-modal for
the truncated model, with peaks at M5.2 and M5.9 (Figure 6). Unlike with Marsan et al. (2017), the peak at lower
magnitude for the truncated model is more probable than the one at larger magnitude. The most probable $M_{max}$
for both models are slightly shifted to lower magnitudes than the values estimated using Marsan et al. (2017)
methodology, but the width of the PDFs appears unchanged to first order. The resulting marginal probabilities
$P(\tau \mid M_w = 5.9)$ and $P(\tau \mid M_w = 5.8)$ for the tapered and truncated models both peak at ~8,000 yrs.
**5.2    Source of seismicity**
We initially selected earthquakes within a 10 km buffer zone around the faults to reflect the spatial strain pattern
of a vertical fault blocked down to a depth of 10 km. Nevertheless, the locking depth could potentially be deeper,
down to ~18 km as suggested in Section 3.1. In this respect, we also provide results if events are selected within
20 km of the faults (Figures S16 and S17). Under these conditions, the seismicity rates of the observational
earthquake catalogs are higher and constrain the long-term seismicity models to cases that produce higher moment
release rate. $P_{M_{max}}$ thus favours events with a lower magnitude than the one using events within 10 km (Figure 5;
Section 4). The tapered model peaks at $M_w 5.9$, instead of 6.1, while the truncated model peaks twice at $M_w 5.2$
and 5.8, in a similar manner to the reference scenario in Section 4, except that the peak at $M_w 5.2$ is now the most
probable.
However, current seismicity in the URG is seemingly diffuse and it is difficult to associate it with a fault in
particular (Doubre et al., 2022). On the other hand, geodetic data are not yet able to resolve any tectonic
deformation and thus to evaluate the loading rate of faults (Henrion et al., 2020). Even though the Drouet et al.
(2020) catalog, based on FCAT-17 catalog, is supposedly devoid of anthropic seismicity (Cara et al., 2015;
Manchuel et al., 2018), one can then ask whether the current seismicity is totally representative of the undergoing
long-term tectonic processes or presently modulated by surface loads such as the post-glacial rebound (e.g. Craig
et al., 2016), aquifer loads, erosion or incision (e.g. Bettinelli et al., 2008; Steer et al., 2014; Craig et al., 2017). If
so, the assumption that the main driver of seismicity is tectonic loading breaks down and our method used to assess
seismic hazard must be completed by physics-based constraints of such transient stress release (Calais et al., 2016).
Distinguishing seismic sources triggered by tectonic loading from other driven forces is an extremely difficult
task. The earthquake catalog contribution (Section 2.3) might then not be appropriate.
Additionally, the magnitudes of historical events from the FCAT-17 catalog (before the 1960s), and thus the ones
from Drouet et al. (2020), seem to be overestimated (or the instrumental events have underestimated magnitudes
even though it seems less probable) and a bias of the MFD is thus expected (Beauval and Bard, 2022; Doubre et
al., 2022). For the URG case, 3 bins out of 7 of the observed MFD are estimated from the instrumental period. The
bins estimated from the historical period have thus slightly more weight in the catalog constraint (Section 2.3).
We test an alternative constraint inferring that the possible magnitude and frequency of $M_{max}$ must be consistent
with the observed largest event over the observation period (~146 yrs), meaning that it has to be larger than or
equal to the known largest event while the return period of the largest event cannot be significantly shorter than
the observation period (Approach 2 from Michel et al., 2018). This constraint is equivalent to considering that no
earthquakes with a magnitude greater than the largest event in the observation period occurred during the time
period covered by the observed catalog. Theoretically, this constraint imposes a lower bound on $M_{max}$ and its
recurrence time. The results obtained using this constraint together with the moment budget and scaling law ones
are shown in Figure 7. Since $M_{max}$ frequency differs for the tapered and truncated models, the new constraint
imposes different lower bounds for the two models. The truncated model rejects scenarios with $M_{max}$ below $M_w 5.5$
more strongly. $P_b$ is not constrained by the observed seismicity catalog but higher values of the b-value seem
slightly more probable (inset in Figure 7). The marginal probabilities $P(\tau \mid M_w = 5.9)$ and $P(\tau \mid M_w = 6.3)$ for
the tapered and truncated models have peaks at ~12,500 yrs and ~63,000 yrs, respectively.

**5.3    Strike slip component**
In this study, as well as in Chartier et al. (2017), we assume solely along-dip displacement since it is the only
published neo-tectonic information available. Nevertheless, recent paleo-seismological data on the Black Forest
fault near Karlsruhe (north of our study area) suggest 5.9 m of cumulative strike-slip, in contrast to 1.2 m of
cumulative vertical slip, over the last 5.9 kyrs (Pena-Castellnou et al., 2023). Those displacements seem to be
associated with at least three paleo-earthquakes. This suggests (1) that the Black Forest fault has been active during
the Quaternary period and that (2) strike-slip might be predominant. The ratio between strike- and dip-slip from
the Black Forest event would be then equal to 4.8. We thus test a scenario where the Black Forest fault is associated
with a maximum vertical slip deficit rate of 0.18 mm/yr, as proposed by Jomard et al. (2017), and where we
multiply the maximum slip deficit rate of all faults considered by 4.8. The results and the revised MDR for each
fault are shown in Figures 8 and S18. $P_{M_{max}}$ peaks at $M_w 6.8$ and $M_w 6.6$ for the tapered and truncated models,
respectively. They are associated with the marginal probabilities $P(\tau \mid M_w = 6.8)$ and $P(\tau \mid M_w = 6.6)$ that both
peak at ~16,000 yrs for the tapered and truncated models. Note that Pena-Castellnou et al. (2023) suggest that
earthquakes of potentially $M_w 6.5$ occurred north of our study area. $P_b$ peaks at 0.7 for both the tapered and
truncated models, thus at lower values than taking into account the vertical-slip component alone.
The previous scenario tested (Figure 8) takes two more faults (i.e. Weinstetten and Lehen-Schonberg faults) into
account than in Chartier et al. (2017), as these two faults are not present within the BDFA (the French database of
potentially active faults; Jomard et al., 2017). The results obtained by selecting faults as defined by Chartier et al.
(2017) and applying the strike slip assumption are provided in Figure S19. $P_{M_{max}}$ peaks at $M_w 6.7$ and $M_w 6.6$ for
the tapered and truncated models, respectively, very similar to the scenario taking all four faults, as the moment
deficit rate is dominated by the Rhine River and Black Forest faults. Note that the marginal probabilities $P(\tau \mid M_w)$
and $P(\tau_{max} \mid M_{max})$ seem to get more noisy, likely due to the shape of the MDR PDF which skews heavily towards
zero (black line in Figure S18.e).
**5.4    Multi-segment rupture**
In this study we assume that all faults can rupture simultaneously. Nevertheless, the Black Forest Fault is initially
taken as inactive, and the traces of the Weinstetten and Lehen-Schonberg faults are separated by at least 7.9 km.
According to Wesnousky (2006), multi-segment ruptures are associated with low probability when the inter
segment distance exceeds 5 km. Consequently, the seismogenic potential scenario from Section 4 would be an
overestimation. On the other hand, according to Castellnou et al., 2022, the Black Forest Fault is in fact active and
seismogenic, and could be assumed to rupture with other faults. Additional structures might actually link all the
faults together (e.g. Lutz and Cleintuar, 1999; Bertrand et al., 2006; Rotstein and Schaming, 2011). In this case,
the seismogenic potential scenario from Section 4 would be interpreted as an underestimation.
Finally, we only consider the faults within a finite zone, which controls the total seismogenic area of the faults (i.e.
the moment-area scaling law effect), whereas the faults continue northwards and southwards to a lesser extent.
According to Weng and Yang (2017), the aspect ratio (width to length ratio of a rupture) of dip-slip events barely
reaches beyond 8. Taking a seismogenic width of 18 km (our maximum estimate), the maximum length of
earthquakes would then be 144 km, while the full length of the URG faults considered, including the Black Forest
fault, is ~250 km (~160 km if the Black Forest fault is not included). The rupture of all the faults would then be
unlikely. On the other hand, strike-slip events do not seem to be capped by any aspect ratio (Weng and Yang,
2017), so $M_w$>7.5 events cannot be excluded in this context.

## 6   CONCLUSION

In this study, we investigate the seismogenic potential of the south-eastern URG, building on the work byChartier
et al. (2017). Based on a complex fault network (Nivière et al., 2008), we evaluate scenarios that have not been
accounted for previously, exploring uncertainties on $M_{max}$, its recurrence time, the $b$-value, and the moment
released aseismically or through aftershocks (see Table 2 for a summary of the results considering the different
scenarios). Uncertainties for the MDR, the observed MFD, and the moment-area scaling law are also explored.
Given the four faults considered, and the scenario in which the Black Forest fault is no longer active but where the
other faults can still rupture simultaneously, the $M_{max}$ maximum probability is estimated at $M_w$6.1 and $M_w$5.8
using the tapered or the truncated seismicity models respectively. Nevertheless, $P_{M_{max}}$ for the truncated model has
a second peak at $M_w$5.2 and the recurrence time of events of such magnitude (not only $M_{max}$), $P(\tau \mid M_w = 5.2) \sim$
2,000 yrs, is much shorter than the one estimated using the main peak, $P(\tau \mid M_w = 5.8) \sim$ 10,000 yrs. Again
considering the scenario excluding the Black Forest fault, there is a 99% probability that $M_{max}$ is less than 7.3
using either the tapered or truncated models. In contrast, when strike-slip kinematics are considered as described
in Section 5.3 and the Black Forest Fault is taken into account, there is a 99% probability that $M_{max}$ is less than
7.6 and 7.5 for the tapered and truncated models, respectively. This is our preferred scenario as it is based on recent
findings for strike-slip mechanisms, although the assumptions made in this analysis are debatable (i.e. strike-
slip/dip-slip ratio evaluated on a fault just north of our zone of study and applied to all faults; Section 5.3).
It should be noted that seismic hazard studies often place an upper bound on the values of $M_{max}$ considered. In
the case of the URG, studies that use varying approaches to ours, have yielded values comparable to, or marginally
lower than the 99th percentile of $P_{M_{max}}$ of our strike-slip scenario (e.g. M7.4, M 7.1 and M7.5 for Grunthal et al.,
2018, Drouet et al., 2020, and Danciu et al., 2021, respectively).
In any case, within this study, strong assumptions still had to be made that certainly affected the results. It includes
the methodology used to decluster the earthquake catalogs, determining whether it is wise to compare the loading
rate of each fault with seismicity, opting to only consider the dip-slip component despite the fact that strike-slip is
highly probable, covering the possibility of multi-segment ruptures and even the choice of the faults to be
considered. Further work, from paleo-seismology, seismic reflection, geodesy, or earthquake relocation is needed
to obtain more information on the structures tectonically involved and their associated loading rates, and to better
constrain the URG seismic hazard.
**7 DATA AVAILABILITY**
The data used in this study are available via the publications mentioned in the main text.
**8 CODE AVAILABILITY**
The analyses reported in this paper were done using MATLAB.
**9 AUTHOR CONTRUBUTION**
SM, CD, LB and RJ conceptualized the study. SM performed the formal analysis with the help of JJ concerning
the declustering of the seismicity catalogs. SM prepared and wrote the manuscript with contributions from all co-
authors.
**10 COMPETING STATEMENT**
The authors acknowledge there are no conflicts of interest recorded.
**11 ACKNOWLEDGEMENT**
This study was supported by the LRC Yves Rocard (Laboratoire de Recherche Conventionnée CEA-ENS-
CNRS) and received funding from the European Research Council (ERC) under the European Union's Horizon
2020 research and innovation program (Geo-4D project, grant agreement 758210). RJ acknowledges funding
from the Institut Universitaire de France. The calculations were performed using MATLAB. We thank the
anonymous reviewers who helped us improve our study.

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

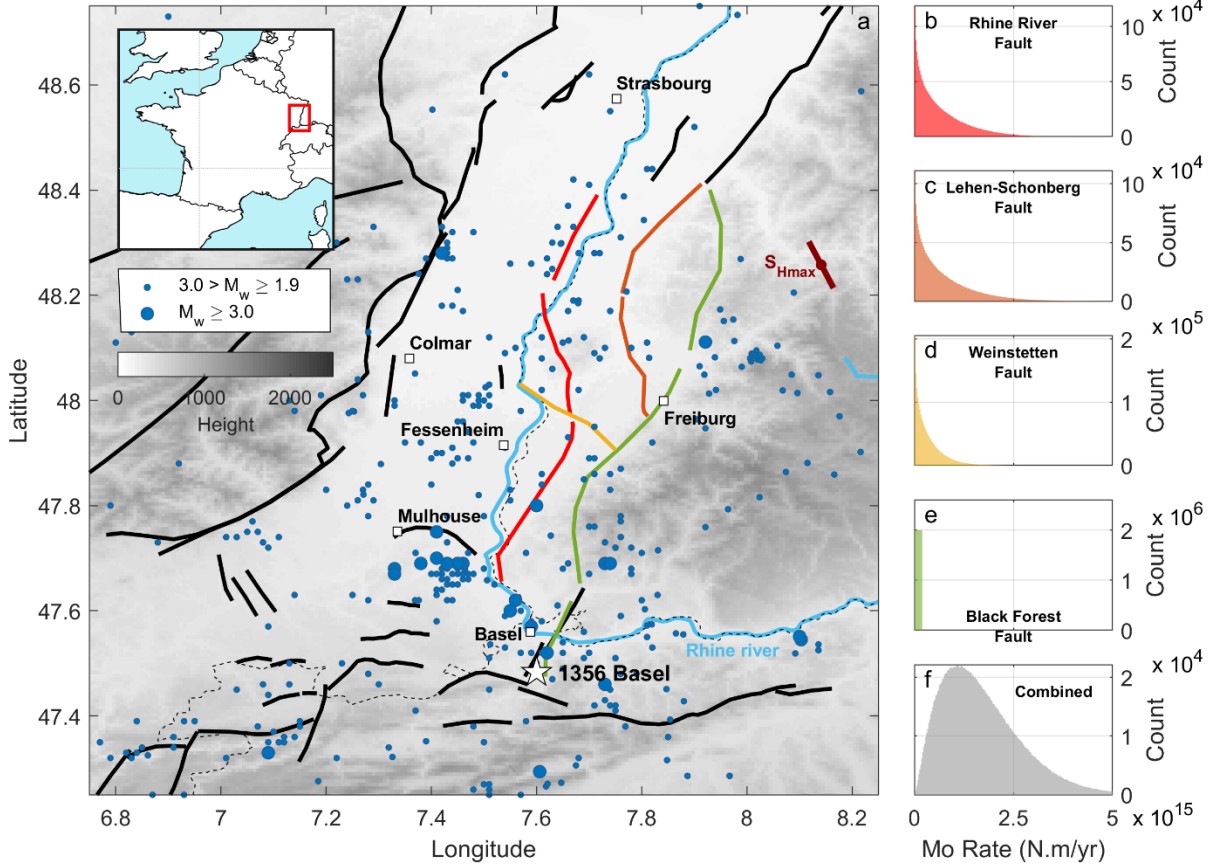


**Figure 1: (a) Regional setting and seismicity of the Upper Rhine Graben (Drouet et al., 2020). Black lines are faults while colored ones are the faults taken into account in this study. The fault network geometry is based on the BDFA database (Jomard et al., 2017) and Nivière et al. (2008). Blue dots are epicenters of $M_w > 2.2$ earthquakes since 1994. The white star indicates the 1356 Basel earthquake (magnitude ranging from M6.5+/-0.5 (Manchuel et al., 2017) to M6.9+/-0.2 (Fäh et al., 2009)). The brown bar indicates the approximate orientation of the maximum horizontal compressional stress ($S_{Hmax}$) (Heidbach et al., 2016, 2018). The thin dashed black line is the border between France and Germany. The nuclear powerplant of Fessenheim and the main cities are indicated by white squares. (b) to (f) Moment deficit rate PDFs (expressed in counts) for each of the four faults considered (colors are indicative of the faults in the left panel), and their combination (in grey).**

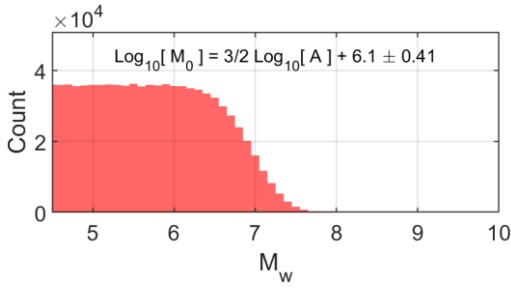

591

**Figure 2: PDF of $M_w$ considering the along-dip moment-area scaling law of earthquakes from Leonard (2010). Note**

**that the area from the Black Forest Fault is not included, as its loading rate is assumed equal to 0 mm/yr.**

594

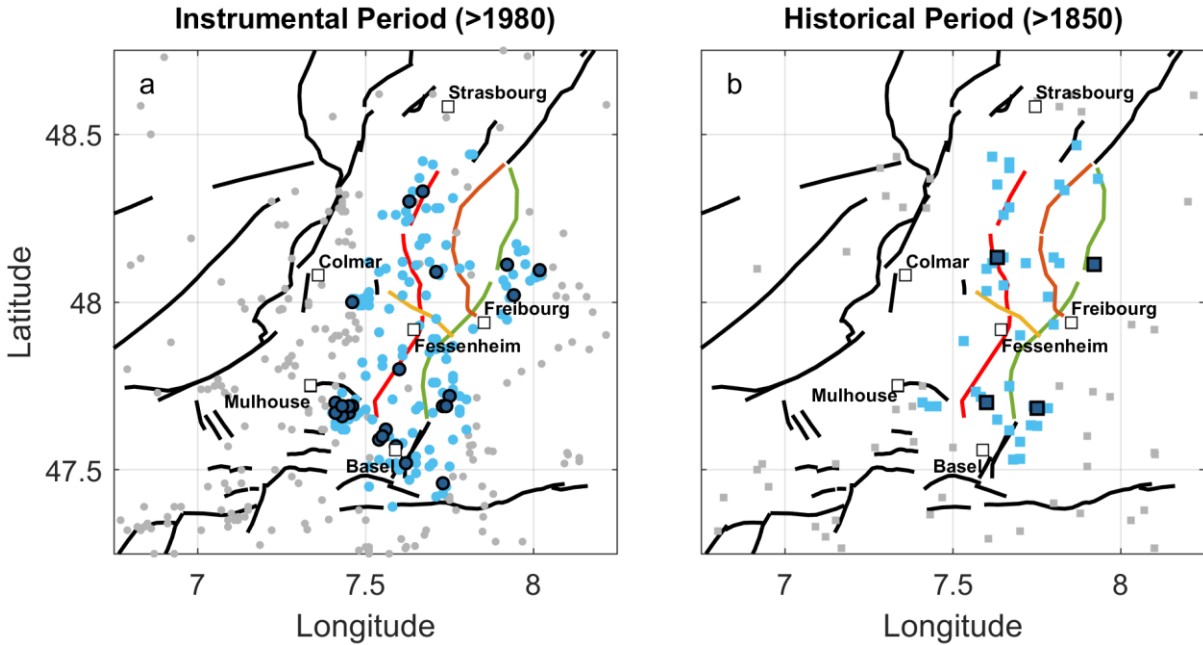

595

**Figure 3: Earthquake selection for the (a) instrumental (>1994) and (b) historical (>1850) periods. Gray dots and**

**squares indicate all earthquakes with $M_c = 2.2$ and 3.2 for the instrumental and historical catalogs, respectively. Light**

**blue dots and squares indicate earthquakes taken into account for the seismogenic potential analysis. Dark blue dots**

**and squares indicate $M_w \geq 2.8$ and 4.3 earthquakes taken into account for the seismogenic potential analysis.**

600

601

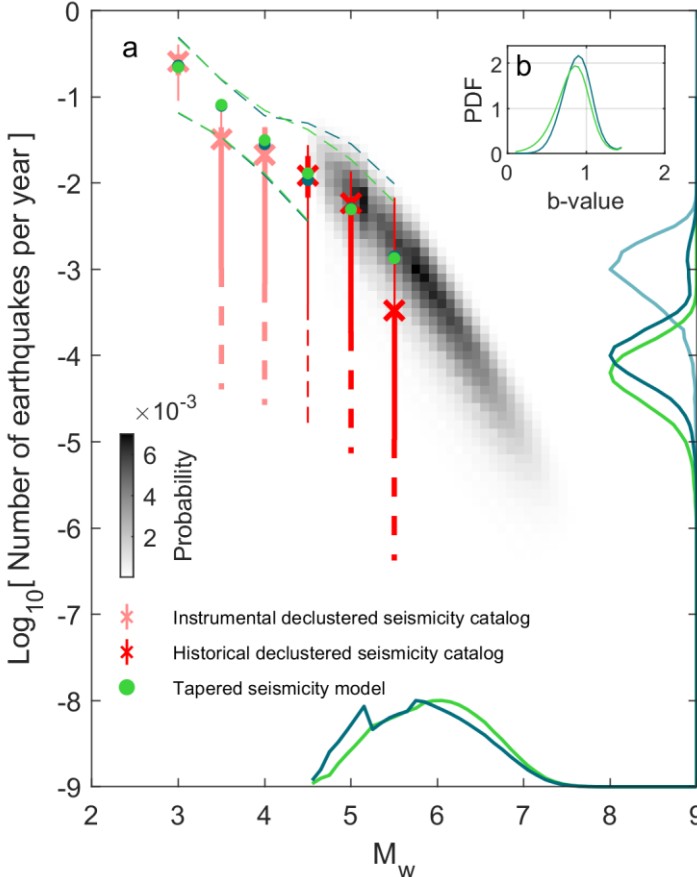

602

**Figure 4: (a) Seismogenic potential of the URG using all constraints: moment budget, observed magnitude-frequency distribution, and moment area scaling law. The rate of occurrence of historical and instrumental earthquakes, within their observation periods, are indicated by red and pink crosses and error bars, respectively. Thick and thin error bars indicate the 15.9-84.1% (1-sigma) and 2.3-97.7% (2-sigma) quantiles of the MFDs. Dashed lines show the spread of possible MFDs for the 2500 catalogs randomly generated to explore uncertainties. The green and blue colors are associated with the tapered and truncated long-term seismicity models. Green and blue dots show the means of the marginal PDF for the long-term seismicity. Dashed green and blue lines indicate the spread of the best 1% seismicity models. The marginal probabilities of $M_{max}$, $P_{M_{max}}$, are indicated by the solid lines on the $M_w$ axis. They have been normalized so that their amplitude is equal to one instead of 0.60 and 0.59 for the tapered and truncated models, respectively.  Green and dark blue lines on the earthquake frequency axis indicate the probability of the rate of events, $\tau$, with magnitude $M_w = M_{Mode}$, thus $P(\tau \mid M_w = M_{Mode})$, with $M_{Mode}$=6.1 and 5.8 for the tapered and truncated models, respectively, considering all magnitudes in the seismicity models and not only the recurrence rate of $M_{max}$. They have also been normalized and their peaks were initially at 1.13 and 1.17 for the tapered and truncated models, respectively. The light blue line on the earthquake frequency axis indicates $P(\tau_{max} \mid M_{max} = 5.8)$ (for the truncated seismicity model only) and is normalized so that its amplitude equals one instead of 1.19. Note that the seismicity MFDs shown in the figure are not in the cumulative form. (b) Marginal probability of the b-value.**

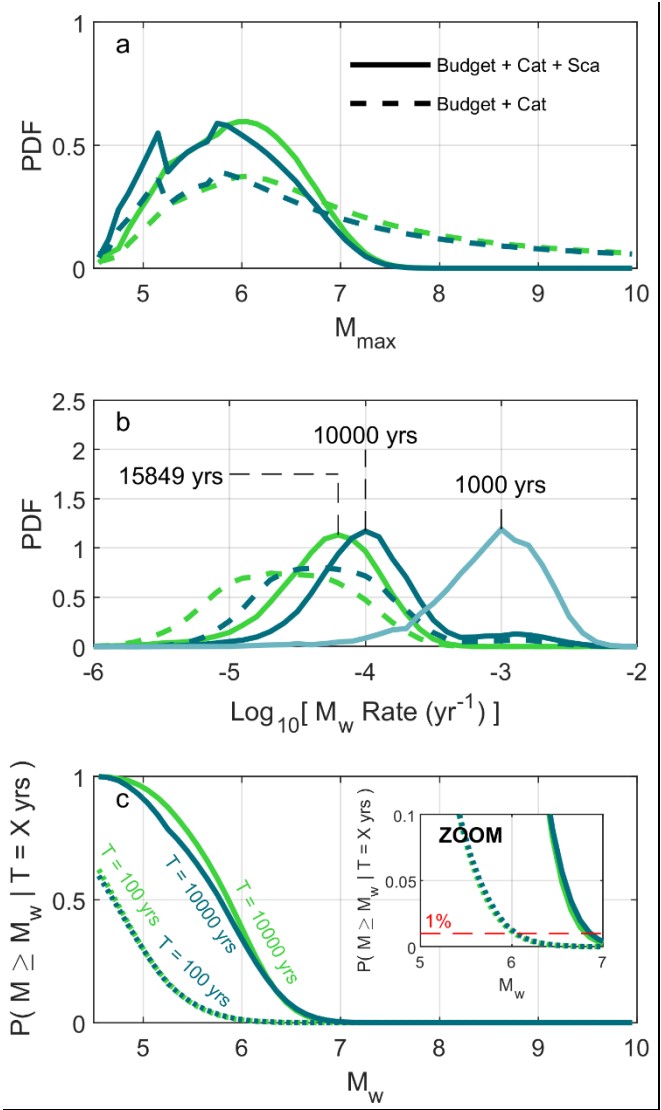

619

Figure 5: (a) Evolution of the marginal PDF of $M_{max}$ when adding the moment-area scaling law constraint. The green

and blue colors in the figure are associated with the tapered and truncated long-term seismicity models. (b) Same as (a)

but for the marginal PDF of the recurrence time of events: $P(\tau \mid M_w = 6.1)$ and $P(\tau \mid M_w = 5.8)$ for the tapered and

truncated models (dark blue and green lines), respectively, and $P(\tau_{max} \mid M_{max} = 5.8)$ shown only for the truncated

model (solid light blue line). (c) Probability of occurrence of earthquakes with a magnitude larger than $M_w$ over a period

of X yrs. We show the probability of occurrence of such events for the 100 yrs and 10,000 yrs time periods. In (a), (b)

and (c), dotted lines represent the marginal PDFs considering both the moment budget and seismicity catalog constraint,

the dashed lines indicate the PDFs when the earthquake scaling constraint is added. The inset in (c) is a zoom of the

panel. The 1% probability of exceedance over a time period of 100 yrs is a typical order of magnitude for nuclear

applications in France.

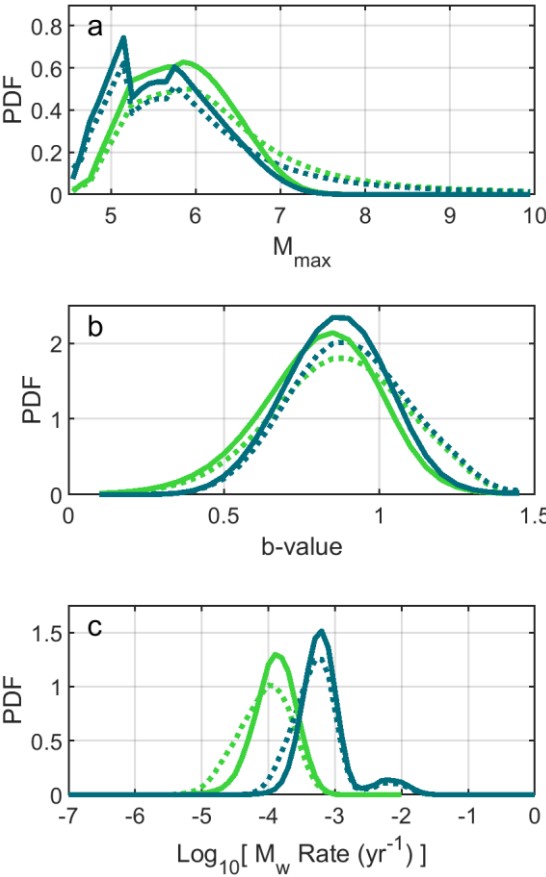

630

**Figure 6: Results using the declustering method from Zaliapin and Ben-Zion (2013) instead of Marsan et al. (2017) (Text S2). In this scenario, no probabilities of events to be mainshocks are defined. (a) $M_{max}$ PDF. (b) b-value PDF. (c) $P(\tau \mid M_w = M_{Mode})$ PDF. Solid lines correspond to the results using all constraints while the dotted lines only use the moment budget and earthquake catalog constraints. Green and blue lines correspond to the tapered and truncated models, respectively. The results shown here are the ones taking a b-value equal to 1 for Zaliapin and Ben-Zion (2013) declustering method. The results for b-values of 0.5 and 1.5 are also shown in Figure S15 and are relatively similar to the ones obtained using a b-value of 1.0.**

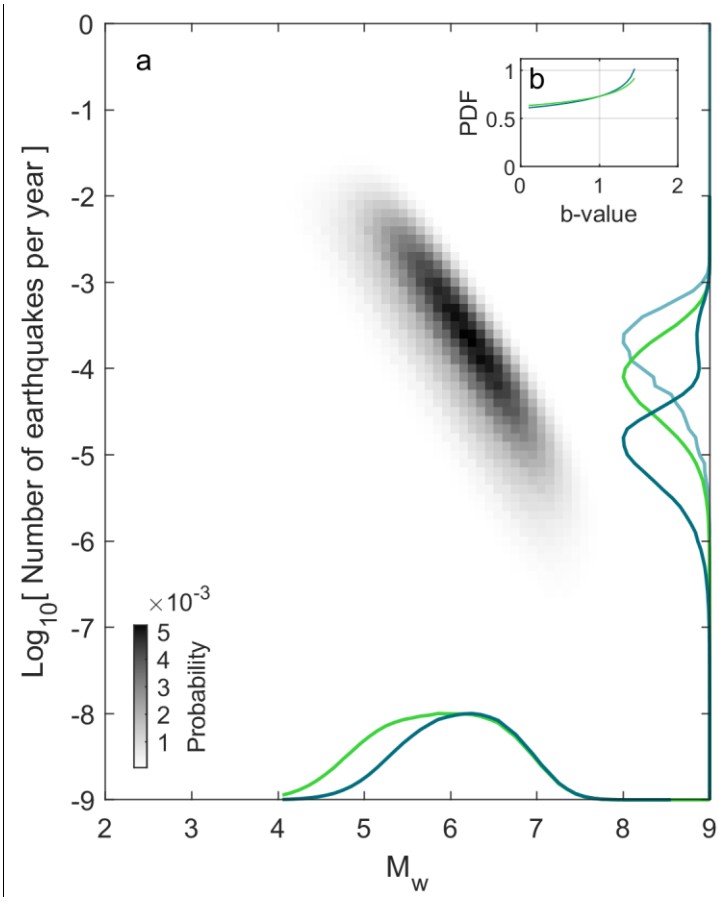

638

**Figure 7: Same as Figure 4 but only considering the constraints for the moment budget, the moment-area scaling law,**

**and the one on $M_{max}$ frequency considering the time period of the catalog (which serves as a lower bound constraint**

**for $M_{max}$; Section 5.2; Approach 2 from Michel et al., 2018). The marginal probabilities $P_{M_{max}}$ have been normalized so**

**that their amplitude is equal to one instead of 0.46 and 0.58 for the tapered and truncated models, respectively. The**

**same is true for $P(\tau \mid M_w = M_{Mode})$ which were initially of 0.85 and 0.81 of amplitude, and $P(\tau_{max} \mid M_{max} = 6.3)$ (for**

**the truncated seismicity model only) which peaked at an amplitude of 0.85.**

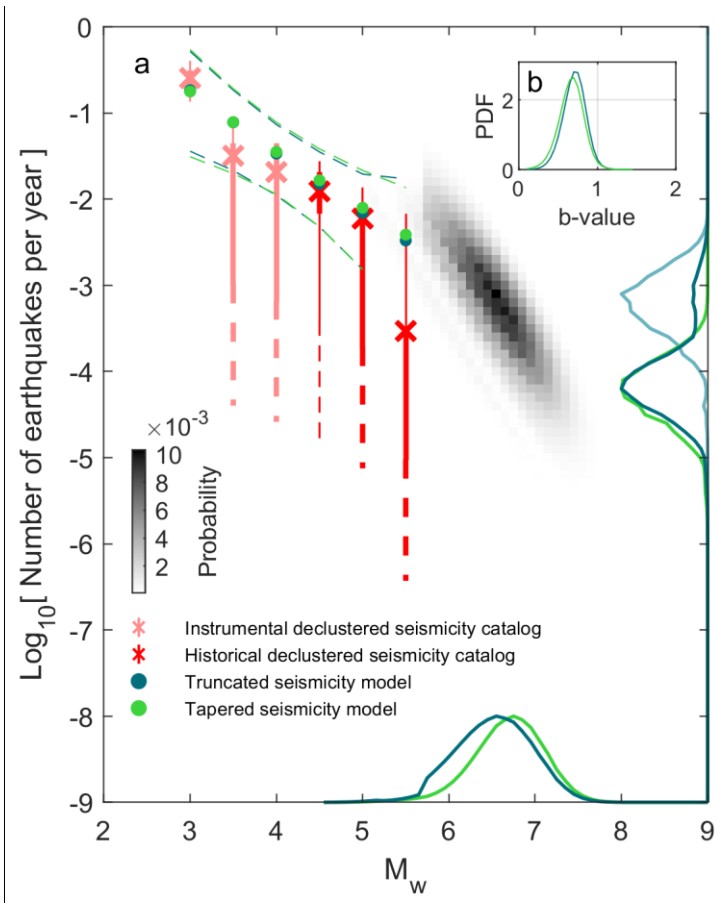


**Figure 8: Same as Figure 2 but considering a strike-slip slip rate component equivalent to 4.8 times the dip-slip estimate,**


**and assuming the Black Forest Fault maximum long-term vertical slip rate is 0.18 mm/yr (as proposed by Jomard et**


**al., 2017). Leonard et al.'s (2010) strike-slip moment-area scaling law is used here for the scaling law constraint, even**


**though it is very similar to the dip-slip version. The marginal probabilities $P_{M_{max}}$ have been normalized so that their**


**amplitude is equal to one instead of 1.02 and 0.88 for the tapered and truncated models respectively. The same is true**


**for $P(\tau \mid M_w = M_{Mode})$ which were initially of 1.15 and 1.13 of amplitude, and $P(\tau_{max} \mid M_{max} = 6.6)$ (for the truncated**


**seismicity model only) which peaked at an amplitude of 1.17.**



**Table 1: Fault parameters. $\mathcal{U}$ and $\mathcal{N}$ stands for uniform and normal distribution. The PDFs of each of these parameters**
**and the resulting moment deficit rate for each fault are shown in Figure S3 to S6.**

| Fault Name | Segment Name (from BDFA) | Dip (°) | Length (km) | Slip-Rate (mm/yr) | Seismogenic zone down-dip extent (km) | Evaporite layer thickness (km) |
|---|---|---|---|---|---|---|
| Rhine River Fault | FRR-1 | $\mathcal{U}(50,80)$ | $\mathcal{N}(35,2)$ | $\mathcal{U}(0,0.07)$ | (1) Uniform from 0 to 6 km in depth. | $\mathcal{U}(0,2)$ |
| | FRR-2 | $\mathcal{U}(50,80)$ | $\mathcal{N}(25,2)$ | | | |
| | FRR-3 | $\mathcal{U}(55,85)$ | $\mathcal{N}(20,2)$ | | (2) Linearly decreasing | |
| Black Forest Fault | FFN-1 | $\mathcal{U}(35,75)$ | $\mathcal{N}(20,5)$ | 0 | from 6 to 18 km depth. | |
| | FFN-2 | $\mathcal{U}(40,80)$ | $\mathcal{N}(50,2)$ | | | |
| | FFN-3 | $\mathcal{U}(35,75)$ | $\mathcal{N}(35,2)$ | | | |
| Lehen-Schonberg | | $\mathcal{U}(40,80)$ | $\mathcal{N}(54,2)$ | $\mathcal{U}(0,0.1)$ | Does not apply to the Black Forest Fault as its loading rate is assumed equal to 0 mm/yr | |
| Weinstetten | | $\mathcal{U}(40,80)$ | $\mathcal{N}(15,2)$ | $\mathcal{U}(0,0.17)$ | | |




**Table 2: Summary of the results considering the different scenarios tested from section 4 to 5.3.**

| Scenarios | Modes of $M_{max}$ | 99% probability that $M_{max}$ is below magnitude $M_w$ | Mode of $P(\tau \mid M_w = M_{Mode})$ |
|---|---|---|---|
| Rhine River Fault + Lehen-Schonberg Fault + Weinstetten Fault<br><br>*Dip-Slip Only Marsan et al. (2017) Declus.*<br><br>(Section 4 / Fig. 4 and 5) | Tapered Model $M_w$ 6.1<br><br>Truncated Model $M_w$ 5.2 and 5.8 | Tapered Model $M_w$ 7.3<br><br>Truncated Model $M_w$ 7.3 | Tapered Model $\tau$ =16,000 yrs<br><br>Truncated Model $\tau$ = 2,000 and 10,000 yrs |
| Rhine River Fault + Lehen-Schonberg Fault + Weinstetten Fault<br><br>*Dip-Slip Only Zaliapin and Ben-Zion (2013) Declus.*<br><br>(Section 5.1 / Fig. 6) | Tapered Model $M_w$ 5.9<br><br>Truncated Model $M_w$ 5.2 and 5.8 | Tapered Model $M_w$ 7.2<br><br>Truncated Model $M_w$ 7.1 | Tapered Model $\tau$ =8,000 yrs<br><br>Truncated Model $\tau$ = 1,600 and 8,000 yrs |
| Rhine River Fault + Lehen-Schonberg Fault + Weinstetten Fault<br><br>*Dip-Slip Only Marsan et al. (2017) Declus. Loose catalog constraint (Approach 2 from Michel et al., 2018)*<br><br>(Section 5.2 / Fig. 7) | Tapered Model $M_w$ 5.9<br><br>Truncated Model $M_w$ 6.3 | Tapered Model $M_w$ 7.4<br><br>Truncated Model $M_w$ 7.4 | Tapered Model $\tau$ =12,500 yrs<br><br>Truncated Model $\tau$ = 63,000 yrs |
| Rhine River Fault + Lehen-Schonberg Fault + Weinstetten Fault + Black Forest Fault<br><br>*Strike- and Dip-Slip Marsan et al. (2017) Declus.*<br><br>(Section 5.3 / Fig. 8) | Tapered Model $M_w$ 6.8<br><br>Truncated Model $M_w$ 6.6 | Tapered Model $M_w$ 7.6<br><br>Truncated Model $M_w$ 7.5 | Tapered Model $\tau$ =16,000 yrs<br><br>Truncated Model $\tau$ = 16,000 yrs |

