# Peer review of "Update of the Seismogenic Potential of the Upper Rhine 1 Graben Southern Region 2"

_EGUsphere, 2023_

## Author Comment (AC1)

REBUTTAL LETTER

We thank the reviewers for their comments, which helped clarify our study. We detail below the changes made to address their comments.

**Reviewer #1**

The article under review covers an interesting topic improving the results of existing literature. While the article is clearly written, in general, some aspects need to be addressed before publication.
General comments:

(1) Reviewer #1

- While the article is generally linearly written and easy to follow, there are a few instances where some concepts are left hanging. A clear example is the shear modulus introduced at line 113 but whose value is defined only at line 194. While the fragmentation of the information may be inevitable due to the number of parameters present in the analysis, referencing the section where the parameters are discussed at length could benefit the readability.

Michel et al.

Thank you for this comment. We now warn the reader at the beginning of Section 2 that the information about the data and their uncertainties are discussed actually in Section 3.

"The data and associated uncertainties used for the constraints are discussed in the following section (i.e. Section 3).''

(2) Reviewer #1

- Some choices introduced as arbitrary could be better motivated (e.g. the linearity of the taper in line 183).

Michel et al.

We changed the sentence to:

"The linearity of the taper implies that the position of the fault's transition to a fully rate-strengthening behavior (>350-450°C) has a uniform probability to fall between 6 km (shallowest position of the 350°C isotherm according to Figure S2) and 18 km depth (deepest position of the 450°C isotherm; Figure S2). "

- The results following the analysis using the tapered and truncated seismicity model are presented and compared but they should be discussed more critically, showing which model should be considered for the following studies. Could the results from the two models be combined? How this would affect the results?

Michel et al.

We prefer to not choose between the truncated or tapered model since we do not know which one is more representative of the region's seismicity. In addition, they are equivalent in terms of model complexity and we cannot rely on Occam's razor to chose between models. Nevertheless, we show now in the supplement a figure similar to Figure 5 but with both models combined (Figure R1 and S12). The PDF of $M_{\text{max}}$ has a main peak at 5.9 and a smaller peak at 5.2, which originates from the truncated model. $P(\tau \mid M_w = 5.9)$ peaks instead at ~13 000 yrs.

[Figure]

[Figure]

**Figure R1**: Same as Fig. 5 but using mixture distribution from the tapered and truncated model. (a) Evolution of the marginal PDF of $M_{\text{max}}$ when adding the moment-area scaling law constraint. (b) Same as (a) but for the marginal PDF of the recurrence time of events: $P(\tau \mid M_w = 5.9)$.

- While already done for some of the assumptions made, a systematic analysis of the results concerning the different parameters could improve the quality of the paper. How the variation of one (or more) parameter affects the results and, more importantly, how this should shape the uncertainty?

Michel et al.

We now show in the supplement the correlation between $M_{max}$, the moment deficit rate, the $b$-value and $\alpha_s$, for both the tapered and truncated model (Figures R2/S10 and R3/S11). We discuss about the correlation between the parameters in Section 4:

"The correlations between $M_{max}$, the moment deficit rate, the $b$-value, and $\alpha_s$, for both the tapered and truncated models but without the scaling law constraint, are shown in Figures S10 and S11. For both models, probable $M_{max}$ increases with increasing $b$-value (Figure S10.a and S11.a), highlighting strong interdependency between the two parameters. Raising the moment deficit rate will control the minimum probable $M_{max}$ (Figures S10.b and S11.b) but will also tend to exclude scenarios with a high b-value (>1.25; Figures S10.f and S11.f). While other trends are expected between parameters, they seem less visible, likely due to the uncertainties of the parameters explored. "

[Figure]

**Figure R2**: Correlation between correlation between $M_{max}$, the moment deficit rate (MDR), the $b$-value, and $\alpha_s$, for the tapered model without the scaling law constraint.

[Figure]

**Figure R3**: Correlation between correlation between $M_{max}$, the moment deficit rate (MDR), the $b$-value, and $\alpha_s$, for the truncated model without the scaling law constraint.

(5) Reviewer #1

- Figures 4, 7, and 8 (as well as S13 and S15) could benefit from the addition of some sort of scale for the PDFs (even though I understand the difficulty due to the figures being already packed with information).

Michel et al.

We apologize to have forgotten to mention it in the caption of those figures. The marginal PDFs on the x- and y-axis are actually normalized by their maximum value and are thus represented with an amplitude of 1. This normalization allows a cleaner presentation on the figure. We keep this format but now refer to this rescaling and give the value of the rescaling for each PDFs in the caption.

(6) Reviewer #1

Targeted comments:

- Is the concept of α_s original or is it taken from previous studies? If the former applies, it should be discussed in more detail; if the latter applies, you should provide some references.

Michel et al.

We now reference Avouac (2015) in Section 2.1 in the manuscript.

(7) Reviewer #1

- In section 2.4 some references are needed relative to how to estimate the parameters from P_SM, as mentioned. In the same section, the variable P_barrier is introduced without being defined.

Michel et al.

We added in Section 2.4 the sentence: "The evaluation of the parameters to estimate $P_{SM}$ are discussed in Section 3". P_barrier is actually a typo and has been removed.

(8) Reviewer #1

- The notation used in line 231 to define the Gaussian distribution N(90%, 25%) can be unclear. Since it refers to a parameter, I suggest changing it to N(0.9, 0.25).

Michel et al.

We changed the notation accordingly.

(9) Reviewer #1

- Section 5.1 provide a clear picture of the variability in the results due to different declustering methods but lacks a concrete discussion motivating how the selection of the algorithm used in the main analysis has been informed. Furthermore, it should be discussed how the results from different declustering algorithms should shape the uncertainty.

Michel et al.

We used in the analysis of reference (Section 4) the declustering methodology from Marsan et al. (2017) because it allows to evaluate the probability of an event to be background seismicity (mentioned in section 3.2). The methodology from Zaliapin and Ben-Zion et al. (2013) does not provide directly such probabilities (it's either background seismicity or not). Although we might be able to modify the Zaliapin & Ben Zion approach to associate a probability to each event, this is clearly not in the scope of this study. In overall, the width of the PDFs using Zaliapin and Ben-Zion (2013) methodology are to first order similar to the results from Marsan et al. (2017).

(10) Reviewer #1

- While the concept in line 374 is clear, the length of future time series (centuries?) needed to improve the results doesn't provide any further information for future studies.

Michel et al.

We removed the sentence:

"Longer time series on all the fields mentioned above might also help in this matter."

(11) Reviewer #1

I also advise the authors to proof-reading the manuscript and the supplementary material: while no major need to be pointed out, there are a few grammatical errors and typos (e.g., the references in the Supplementary material are addressed as "Bibliographie").

Michel et al.

The manuscript has now been read by a native English speaker and we hope it has minimized the number of errors/typos.

---

## Author Comment (AC2)

REBUTTAL LETTER

We thank the reviewers for their comments, which helped clarify our study. We detail below the changes made to address their comments.

**Reviewer #2**

The article focuses on the Upper Rhine Graben's seismogenic potentials. Significance of the article is to improve the modeling of the seismogenic zones by including various parameters related to the fault modeling and seismicity of the region. In this regards, the article helps on constructing a better seismic hazard model for the region.
The paper has a good quality but there are some aspects of the paper that requires improvements. They are given in below.
**Major comments:**

(1) Reviewer #2

- It would be better to provide a more expanded information about the truncated and tapered models. Maybe it is a very well known and used methods in the subject but it would help reader to understand the basic consent without reading another paper(s).

Michel et al.

We now include in the supplement a figure illustrating the two models (Figure R4 / S1).

[Figure]

[Figure]

**Figure R4**: Illustration of the tapered and the truncated models in their (a) cumulative form and (b) non-cumulative form. $M_{max}$, the recurrence time of events of a certain magnitude ($\tau_c$), and the b-value indicate the three parameters that control those models.

**(2) Reviewer #2**

- Magnitude estimations are given in the second decimals which requires a great knowledge of the almost all the parameters in the study. However, there are lot's of uncertainties in both data (eg. lack of knowledge about seismogenic depth) and method (eg. using a $m_0$ and $A_{eq}$ scaling law which has its own standard deviations). In this regard, providing earthquake magnitudes in second decimals as if there is such a good resolution in the study is not realistic.

Michel et al.

We agree and rounded them.

**(3) Reviewer #2**

- In the thermal gradient part of the Section 3.1 the thermal features are expected to increase linearly with depth. Is there any reference to justify this linear gradient in depth? Neither Freymark et al. (2017) nor Guillou-Frottier et al. (2013) has any model for the depths that are deeper than the drilling depths. Even though Freymark et al. (2017) had a model for density variations, in thermal data the study limits itself except for the Fig. 10F which does not provide a detailed model for the region.

Michel et al.

Since we do not have enough data to constrain this point, we assume a homogeneous medium and steady-state conditions, resulting in a linear increase of temperatures with depth. Additionally, to first order, measurements of temperature down to 2 km-depth in the URG (Guillou-Frottier et al., 2013) seems to indicate a linear thermal gradient (Figure S2). We just extrapolated. We now state in Section 3.1 that we assume such linear gradient.

**(4) Reviewer #2**

- In the Section 3.1 seismogenic thickness of the southern par of the URG is defined by using temperature and salt basin assumptions. However, in the paper it is stated that in the southern URG there is seismicity. Can the authors add seismicity for the thickness modeling?

Michel et al.

The depth uncertainties in our seismicity catalog are unfortunately very large, if not undefined, and we prefer not to use them to infer the depth of the seismogenic zone. The variable quality of the depth determination is due to both the sparsity of the network at the beginning of the

instrumental period, the lack of stations (records) close to the event epicenters, and to the proximity to the German border, which sometimes requires the mixing of different national seismological networks to reduce the epicentral gaps. We have checked the ISC-EHB catalogue containing well-constrained hypocentral depths (level L1), or the IASPEI Ground Truth GT5 events, but the number of earthquakes in these catalogues is unfortunately too small for our region of interest. Furthermore, we believe that the depth of small-magnitude events, even if well recorded, may be not representative of the depth of larger events.

**(5) Reviewer #2**

- In Section 3.1 and Section 3.3, relation between the section and their effect on the model parameters are explained by referring the variables in sections. It would be good to provide same information for Section 3.2.

Michel et al.

We now changed the beginning of Section 3.2 to:

"To constrain the MFD of the long-term seismicity models with an observational seismicity catalog, as described in Section 2.3, we need to evaluate from the observational catalog the number of events per magnitude bin $n_{obs}^{M_i}$ over a period of time $t_{obs}^{M_i}$ (Section 2.3). We use the earthquake catalog from Drouet et al. (2020). This catalog was built from multiple former catalogs. …"

**(6) Reviewer #2**

- In line 267-268 what is the reason for choosing second peak in truncated model instead of the first one?

Michel et al.

It's the largest peak, thus the mode of the PDF. But indeed we could have done it also for the second peak. Nevertheless, we prefer not to do it as the reading would become cumbersome.

**(7) Reviewer #2**

- In line 149, $P_{barries}$ is not defined.

Michel et al.

Thank you for pointing this out. It's a typo and we removed $P_{barrier}$ from the equation.

**(8) Reviewer #2**

- To sum up the study, a Table can be added into the conclusion part which shows the changing results for changing parameters. They are more or less presented in the text of conclusion. A table show the overall results which helps readers to summarize the study.

Michel et al.

We now added such table in the conclusion (Table 2).

**(9) Reviewer #2**

- As mentioned in Introduction and Figure 5, results of this study can be useful for nuclear power plant constructions. However, in the paper there is no indication of which model is better and should be preferred on hazard assessments. In a scientific point of view, all the variations in results are important but for the engineering perspective there must be a decision to make to choose a number to move forward. In the conclusion there is no such indications.

Michel et al.

We added the following sentence in the conclusion:

"In contrast, when strike-slip kinematics are considered as described in Section 5.3 and the Black Forest Fault is taken into account, there is a 99% probability that $M_{max}$ is less than 7.6 and 7.5 for the tapered and truncated models, respectively. This is our preferred scenario as it is based on recent findings for strike-slip mechanisms, although the assumptions made in this analysis are debatable (i.e. strike-slip/dip-slip ratio evaluated on a fault just north of our zone of study and applied to all faults; Section 5.3). "

**(10) Reviewer #2**
  **Minor details:**

- Line 58 – "w" of M should be subscript to be in agreement with the other $M_w$s.

Michel et al.

We changed the subscripts accordingly.

**(11) Reviewer #2**

- Line 128 – No need to say "according to global earthquake statistics".

Michel et al.

We prefer to keep it.

**(12) Reviewer #2**

- Line 136 – No need to say "finally".

Michel et al.

Done.

**(13) Reviewer #2**

- Line 138 – "since" is a conjunction, so it should always join two clauses.

Michel et al.

Thank you for pointing this typo out. We modified the sentence accordingly:

"Since we consider only mainshocks, we define the likelihood of the observed seismicity catalog …"

**(14) Reviewer #2**

- Line 170 – "aforementioned" can be moved before the "other". "… by the aforementioned other faults".

Michel et al.

Done.

**(15) Reviewer #2**

- Line 231 – What does percentage of the mean and variance of the scaling factor in numbers?

Michel et al.

We apologize for the confusion, we changed the sentence to:

"… we assume the PDF of $\alpha_s$ is a Gaussian distribution with $\mathcal{N}(0.9, 0.25)$ …"

**(16) Reviewer #2**

- Lines 275-278 – Word "thus" used in 3 consecutive sentences. Different wording can be chosen to text more readable.

Michel et al.

We changed to three sentences to:

"In this respect, we also provide results if events are selected within 20 km of the faults (Figures S16 and S17). Under these conditions, the seismicity rates of the observational earthquake catalogs are higher and constrain the long-term seismicity models to cases that produce higher moment release rate

**(17) Reviewer #2**

- Please provide the DOI for all the references.

Michel et al.

Done.

**(18) Reviewer #2**

- In Figures 4, 7-8 it would be better to put a y-axis for $P_{mmax}$ and and tau.

Michel et al.

See Answer to comment *(5) Reviewer #1*.

---

## Referee Report (RR1)

**EGUSPHERE-2023-359**

The manuscript covers an interesting topic, already discussed in literature, providing improvements on previous works.

The manuscript is well-written and provides a good description of the existing literature used as a starting point.

The methodology adopted is clearly explained and the choices performed in defining the parameters used during the analysis are well-motivated.

The results are clearly discussed and well summarized (Table 2 is a nice addition to the manuscript).

A few minor details should be fixed before publication:

- In section 3.3 the notation $U(..., ...)$ is used to define a uniform distribution between two values without being properly introduced. Also, the same notation should be used in section 3.1 when discussing the PDF of the seismogenic down-dip extent for consistency.
- At the end of section 4 the sentence "While other trends are expected between parameters, they seem less visible, likely due to the uncertainties of the parameters explored." is ambiguous as it is not clear if the dependencies among these parameters have been studied or not. If the high uncertainty of these parameters is the motivation not to pursue further analyses it should be stated more clearly.
- Not all figures presenting multiple panels are labelled: I suggest adding a label to each subfigure and properly referencing it in the caption.

---

## Author Response (AR2)

Dear Editor,

We thank the reviewers and have revised our manuscript to take into account their comments. Please find below our answer to the comments.

On behalf of all co-authors,

Sylvain MICHEL

**Reviewer #1**

The manuscript covers an interesting topic, already discussed in literature, providing improvements on previous works. The manuscript is well-written and provides a good description of the existing literature used as a starting point. The methodology adopted is clearly explained and the choices performed in defining the parameters used during the analysis are well-motivated. The results are clearly discussed and well summarized (Table 2 is a nice addition to the manuscript). A few minor details should be fixed before publication:

(1) Reviewer #1
- In section 3.3 the notation U(..., ...) is used to define a uniform distribution between two values without being properly introduced. Also, the same notation should be used in section 3.1 when discussing the PDF of the seismogenic down-dip extent for consistency.

Michel et al.
We now introduce at the beginning of Section 3 the symbol $\mathcal{U}$ and $\mathcal{N}$, and refer also to Table 1 that summarizes the uncertainties taken for each parameter. We modified two sentences in Section 3.1 so that we now use the notation $\mathcal{U}$. However, since this notation has been introduced before, we can keep Section 3.3 as is.

(2) Reviewer #1
- At the end of section 4 the sentence "While other trends are expected between parameters, they seem less visible, likely due to the uncertainties of the parameters explored." is ambiguous as it is not clear if the dependencies among these parameters have been studied or not. If the high uncertainty of these parameters is the motivation not to pursue further analyses it should be stated more clearly.

Michel et al.
We changed the sentence accordingly:
"While other trends are expected between parameters, they seem less visible likely due to the uncertainties of the parameters explored, and we thus do not pursue further analysis between those parameters."

(3) Reviewer #1
- Not all figures presenting multiple panels are labelled: I suggest adding a label to each subfigure and properly referencing it in the caption.

Michel et al.
Done.